# Bon bagay (good stuff): A faith-based outlook on biomedical prevention among Haitians and Haitian Americans

Candice A. Sternberg[1]*, Maurice J. Chery[2], Maika Beauvoir[1], Pepita Jean[1],
Dominique Guillaume[3], Joelle-Ann Joseph[1], Regine Thermy Jean Baptiste[4],
Tyra Montour[5], Valeria Botero[1], Anjalie Geffrard[1], Chantal Napoleon[1], Terese Gelin[1],
Allan Rodriguez[1], John F. P. Bridges[6], Christopher J. Hoffmann[7], Guerda Nicolas[4],
Sannisha K. Dale[8‡], Maria L. Alcaide[1‡]

**1** Department of Medicine, University of Miami Miller School of Medicine, Miami, Florida, United States of America, **2** Department of Public Health Sciences, University of Miami Miller School of Medicine, Miami, Florida, United States of America, **3** Center for Infectious Disease and Nursing Innovation, Johns Hopkins University, Baltimore, Maryland, United States of America, **4** Department of Educational and Psychological Studies, University of Miami, Miami, Florida, United States of America, **5** Department of Health Behavior, Texas A&M University, College Station, Texas, United States of America, **6** Department of Biomedical Informatics and Surgery, Ohio State University, Columbus, Ohio, United States of America, **7** Department of Medicine, Johns Hopkins University, Baltimore, Maryland, United States of America, **8** Department of Psychology, University of Miami, Florida, United States of America

‡ SKD and MLA are joint senior authors on this work.
* c.aurelus@miami.edu

## Abstract

### Background

Miami-Dade, Florida is a key hotspot for new HIV diagnoses. Haitians and Haitian Americans have been disproportionately affected. Churches play a critical role in information delivery in the Haitian community. This study provides an understanding of perceptions regarding Pre-exposure prophylaxis (PrEP) for HIV prevention among key informants.

### Methods

In this qualitative study, focus groups were conducted with Haitian church leaders using snowball sampling. A semi-structured interview guide was used to engage discussions on topics including HIV prevention, PrEP, barriers to engagement in PrEP, and current services provided. Focus groups were audio recorded and transcribed. Thematic analysis was conducted on NVIVO computer software using a general inductive approach.

### Results

Three focus groups were conducted. Twenty-seven (16 women and 11 men) individuals participated, most of whom were born in Haiti (78%) with an average age of 48.

**Data availability statement:** All relevant data are within the manuscript and its Supporting Information files.

**Funding:** This work was supported by the National Institute on Minority Health and Health Disparities, award number P50MD017347-02S1. Dr. Sternberg has also been supported by the University of Miami's Center for HIV and Research in Mental Health (CHARM); P30MH116867 (Developmental-AIDS Research Center) and P30MH133399 (National Institute of Mental Health). Lastly, Dr. Sternberg is supported by the University of Miami Clinical and Translational Science Institute (CTSI) through the National Center for Advancing Translational Sciences, award number KL2TR002737. The funders had no role in study design, data collection and analysis, decision to publish, or preparation of the manuscript.

**Competing interests:** The authors have declared that no competing interests exist.

Eight key themes that emerged from the focus groups included the democratic nature of the churches, stigma, and fear regarding HIV/AIDS, lack of knowledge regarding PrEP, acknowledgment of PrEP benefits, trust, the churches' roles as educators and culturally relevant messaging.

## Conclusion

Churches, in partnership with trusted medical professionals and using culturally relevant messaging, are likely key strategies for increasing PrEP awareness among Haitians in Miami. Organizing health fairs and educational meetings can make churches effective platforms for PrEP awareness, leveraging their role as trusted community institutions.

## Introduction

The Human Immunodeficiency Virus (HIV) epidemic continues to be a public health crisis in South Florida. While the successes of antiretroviral therapies, among other factors, have resulted in a decline of HIV incidence and deaths, incidence remains above target epidemic control goals [1]. Certain states have disproportionally higher rates of HIV prevalence than the national average, including Florida [2]. Within Florida, Miami-Dade has the highest HIV prevalence [3]. Miami-Dade has been identified as one of the 48 hotspots of the HIV epidemic in the U.S. [4].

Moreover, Black, or African American individuals remain disproportionately affected by HIV largely due to structural factors and marginalization [1,5]. In 2021, the Center for Disease Control (CDC) data reports showed that Black or African American individuals, made up 40% of new HIV diagnoses in the U.S. [1]. Data from the Florida Department of Health for Miami-Dade County regarding Haitian-born individuals, showed the prevalence of AIDS among Haitian-born people is 12.9%, and Haitian-born people make up 5.5% of new HIV diagnoses [6]. Of concern, among immigrants to the US, the rate of HIV-related deaths is highest amongst Haitian immigrants at 14.2% [6].

### HIV prevention

One effort to curb the HIV epidemic was the Food and Drug Administration's (FDA) 2012 approval of pre-exposure prophylaxis, commonly known as PrEP [4]. When taken as prescribed, oral PrEP reduces the risk of contracting HIV by 99% [7,8]. In 2023, the CDC reported that only 10% of Black individuals who have indications for PrEP are prescribed the regimen, compared to their Latino or White counterparts, among whom 21% and 82% of individual with indications for PrEP are prescribed a regimen, respectively [9]. Thus, the rate of prescribed PrEP among Haitians and Haitian Americans is also likely to be low.

### Historical context of HIV/AIDS

In addressing HIV prevention within the Haitian community, it is imperative to acknowledge historical contexts that have contributed to stigma, which can impact

engagement in HIV prevention behaviors, including the uptake of PrEP. Early in the HIV epidemic, researchers at the CDC inaccurately generalized Haitians as 'AIDS carriers' [10]. According to Santana & Dancy [11], this blaming compounded existing stereotypes about Haitians, consequentially adding to the list of negative labels that American society placed on Haitians at that time (e.g., boat people). The erroneous framing of Haitians as being AIDS carriers resulted in heightened public stigmatization of the Haitian community, promoting the perception that AIDS was a condition that affected nearly all Haitians [11]. The FDA even prohibited blood donations from Haitians [12], an example of a consequence of this immense stigmatization.

While these events occurred four decades ago, and the CDC later retracted its statement with much less publicity than the original, the subsequent impact of the stigma continues today [11]. The stigmatizing of this population resulted in significant health and social consequences that have amplified health privacy concerns and hindered social acculturation and integration [11,13–15]. This stigma also created additional barriers to care for Haitian individuals living with HIV [16]. It is also important to note HIV's stigma in Haiti, where it is viewed as a consequence of sexual immorality [17]. Hence, these stigmas have intersected so that Haitian immigrants avoid discussions related to HIV/AIDS and sexual and reproductive health in general.

Although PrEP is used for HIV prevention, numerous studies have reported societal misconceptions that it is used for treatment, thereby conveying the stigma of HIV to PrEP users [18–21]). According to Calabrese [19], central to these stereotypes is the unfavorable assumption that PrEP users engage in condomless sex, have more sexual partners, and engage in higher-risk sexual behaviors. These negative perceptions have been widely reported within Black immigrant communities [22–23]. This stigmatizes individuals, keeping them from accessing and engaging in care and thus reducing intervention effectiveness [24].

## Role of black churches

Religion, regardless of the denomination, plays a significant role in the lives of many Haitians and tends to govern and influence their decision-making process. Many Haitians believe that religion is necessary to maintain peace and order and lack of religious values or lack of respect for religion can be detrimental to the country [25]. Furthermore, different beliefs systems can portray HIV in a supernatural light, as a punishment or a curse, which is a belief held by other groups including Black Americans [26–28].

Research suggests that Black American churches are crucial partners in HIV prevention and care efforts although these studies have not examined Haitian churches in particular. Black American churches are well-positioned to increase the reach of HIV prevention interventions and address HIV-related stigma within their communities [29]. Many churches increasingly engage in HIV awareness, screening, and prevention, with most members supporting these activities [29–30]. Black American faith leaders have provided recommendations for reducing racial disparities regarding HIV/AIDS, emphasizing the importance of promoting HIV testing, integrating HIV/AIDS topics into health messaging and sermons, and conducting community outreach and educational sessions for youth [31]. Integrating church-based and public health perspectives has been identified as a key strategy for churches that want to engage in HIV-prevention activities while maintaining their spiritual priorities and accountability [32].

Additionally, recommendations have been made for churches to offer more inclusive HIV prevention efforts by reducing stigma toward LGBTQIA (lesbian, gay, bisexual, transgender, queer, intersex, and asexual communities) individuals and addressing the sexual health needs of all congregants [33].

This study is part of an effort to leverage the church's role in increasing PrEP knowledge and use in the Haitian community in Miami-Dade County, Florida, and surrounding areas. The *Bon Sante* study, looked at factors influencing PrEP and noted that there were multiple opportunities to increase PrEP awareness [16]. One of these opportunities was through working with churches. The purpose of this study was to get a better understanding of HIV prevention in the Haitian community in Miami-Dade County and surrounding areas while examining the role that churches can play in PrEP

delivery. The objective of the focus groups was to obtain information from church leaders regarding (1) the structure of the churches, (2) thoughts on HIV/AIDS, (3) thoughts regarding PrEP, (4) barriers or facilitators of PrEP, (5) role of the church in PrEP delivery, (6) current services provided, and (7) potential interventions to help with PrEP awareness.

## Methods

The Haitian American Professionals Coalition (HAPC) is a group of Haitian professionals who served as the advisory board and provided guidance regarding study measures. Prior to study commencement, the principal investigator (PI) contacted leaders from different churches in Miami-Dade County, FL, and the surrounding areas, through collaborations with the HAPC. The PI proposed the project at this meeting, explained its importance, and determined interest. A qualitative approach was used to carry out study goals. Data was collected from 07/30/23 through 08/26/23. The study team consisted of Haitian American women and men, fluent in English and Haitian Creole. To protect research data and ensure privacy, all participant data was stored in a secured, password protected University of Miami approved cloud platform (Box) and data collection program (RedCap®- Research Electronic Data Capture). All study staff received training regarding participant privacy. A link between ID number and participant's name were kept in a separate file and password protected.

### Recruitment

Community consultants (2 Haitian women with strong ties to their community) and a research associate connected with multiple churches face to face, via telephone and email seeking partnership after an initial Zoom meeting with seven pastors from local churches. Two pastors declined during the initial meeting because they felt uncomfortable with the topic. The first three churches to agree to participate (after consulting with their committees) were recruited to the study. These churches agreed to allow recruitment and focus groups at their institutions. Two churches were in Miami and one in Hollywood, Florida.

Once the partnerships were formed, the snowball method was used to recruit study participants. Leaders of the churches (pastors and committee members, such as leaders of the children's ministry and worship team) were encouraged to identify and invite other potential participants who held leadership positions in the church. The sample size was established using published scientific research protocols in which the recommended number of participants are between 4–12 individuals per focus group [34]. Twenty-eight participants were recruited for this study. However, 1 participant dropped out after the review of the consent. Given the sensitive nature of HIV- related discussions regarding Haitians with church leaders, this was a confidential minimal risk study. The University of Miami Institutional Review Board waived the requirements to obtain signed consent. Verbal consent was obtained from all individuals. The survey did not proceed until clicking yes to consent on the computer tablets. In total twenty-seven participants participated in this study. Participants received $50 for their efforts.

The University of Miami Institutional Review Board approved this study. After the study was completed, the study team returned to all churches to relay the data and obtain feedback regarding the findings.

### Measures

*Demographic Assessment.* Participants were required to complete a demographic survey using RedCap® (Research Electronic Data Capture). Due to unknown literacy levels among the sample, team members assisted participants in completing the survey. The assessment consisted of the following socio-demographic factors: age, gender, education level, religion/ domination, housing arrangement, household income, work/school status, employer affiliation, country of birth, and referral means. Table 1 shows socio-demographic factors.

*Focus Group Guide.* The research team developed a focus group guide in collaboration with the Haitian American Professionals Coalition (HAPC) (the focus group guide is shown in the S1 Appendix). Moderators led discussions in participants' preferred language. They encouraged them to share their knowledge/opinions regarding (1) HIV prevention, (2) PrEP, (3)

**Table 1. Participant Demographics.**

| Variables | N | Percent | Mean | SD |
|---|---|---|---|---|
| **Age** | | | 47.9 | 14.2 |
| **Gender** | | | | |
| Female | 16 | 59.3% | | |
| Male | 11 | 40.7% | | |
| **Education** | | | | |
| 8th grade or lower | 1 | 3.7% | | |
| Some high school | 2 | 7.4% | | |
| Highschool graduate | 2 | 7.4% | | |
| Some college | 8 | 29.6% | | |
| College graduate | 6 | 22.2% | | |
| Some graduate school | 2 | 7.4% | | |
| Graduate school | 6 | 22.2% | | |
| **Religion** | | | | |
| Christian (Non-denominational, Baptist, Protestant, Adventist) | 27 | 100% | | |
| **Housing Arrangement** | | | | |
| Renting | 13 | 48.1% | | |
| Own | 14 | 51.9% | | |
| **Work/ School** | | | | |
| Full time/ Part time work | 4 | 66.7% | | |
| Full time/ Part time school | 18 | 66.7% | | |
| No school nor work | 1 | 3.7% | | |
| On disability/other | 1 | 3.7% | | |
| **Household Income** | | | | |
| >$5,000 | 2 | 7.4% | | |
| $5,000-$24,999 | 2 | 7.4% | | |
| $25,000-49,999 | 4 | 14.8% | | |
| $50,000 or greater | 15 | 55.6% | | |
| I choose not to answer | | | | |
| **Birthplace** | | | | |
| USA | 4 | 14.8% | | |
| Haiti | 21 | 77.8% | | |
| Other | 2 | 7.4% | | |

stigma associated with PrEP, (4) the church's potential role in PrEP delivery, (5) barriers, (6) facilitators, and (7) strategies to distribute PrEP. The focus groups were conducted at three Christian churches in the Greater Miami area. Two focus groups were conducted in Haitian Creole and one in English. Only the research team members and participants were present during discussions. All three sessions were audio recorded with the consent of participants and convened on average for about 1 hour and 30 minutes. Participants were asked if they would like to review the audio recording, but they all declined. De-identified recordings were later sent to a company to be transcribed and translated to English text.

## Data analysis

Thematic analysis was conducted to analyze data utilizing an inductive approach [35]. The PI and two other team members each coded one of the transcripts (3 total) to identify major themes. The three coders then collaboratively developed a preliminary code book. The coders met two times to discuss their codes, emergent themes, and resolve discrepancies

through collective consensus. Once consensus was reached, the final coding structure was reviewed by the PI. Each theme was defined and accompanied by exemplary quotes. Once the coding framework was established, the PI and research associate continued to iteratively analyze each transcript independently using NVivo, a qualitative data analyses program [36]. Kappas below 0.08 were reviewed and discussed by coders. All Kappas were above 0.8 and data saturation was reached. Study team members conducted member checking by presenting results to the churches to ensure interpretations accurately reflect the participants' perspectives [37]. Church leaders provided feedback on the results and shared their thoughts on the project.

## Results

Three focus groups were conducted with a total of 27 participants. Sixteen individuals identified as women, and 11 identified as men. Most (77.8%) of those individuals were born in Haiti. The average age of the participants was 47.9 years old (SD = 14.2). Over 80% of the participants had completed at least some college, and 100% considered themselves Christian. Fifty-one percent (51.9%) reported that their home was owned by them or someone else in the household. More than half (66.7%) of participants worked or were in school full-time and 29.6% had a household income of less than $50,000.

Multiple members of the churches took part in the study, including a youth leader and one who oversees the women's ministry (Focus group 1, Participant 3; Focus group 3, participant 24, respectively). Church leaders expressed that they believed the study findings accurately reflects their views and that PrEP awareness is needed in the Haitian community.

### Structure of church/Objectives

Most of the churches had a democratic structure and committees to make decisions. A common goal for the churches was to educate the community. "*Ideas come from anyone. They are presented to the church at first… then we do all the decisions in the committee then we refer it back to the church to vote. It's pretty much a voting process.*" (Focus group 1, Participant 5).

### Thoughts regarding HIV/AIDS

Participants noted the historical context of HIV and Haitians. "*As Haitians, we can say that we are victimized by HIV because at first, they said that Haitians were the initial bearers of the disease. This caused the marginalization of Haitians. Wherever Haitians were present they were linked to what was called HIV, what they called AIDS. They said that the disease originated from Haitians. Well, this was politics.*" (Focus group 3, Participant 19).

This study identified a high level of stigma regarding HIV. "*Someone who is at church and catches this disease in these kinds of activities are not honest. They are not serious. They have no respect for themselves and for God.*" (Focus group 1, Participant 7). Particularly interesting was that there was a stigma coming from healthcare professionals. "*I would say the stigma still exists even among professionals. Because even sometimes at work, let's say you gave a report to someone of a patient, and you say this person has High5 and they are like Huh. You know. It's like you know, just give me the report.*" (Focus group 1, Participant 2). In addition, some suggested that members of the Haitian community contributed to the stigma regarding HIV. "*We are the reason that they are stigmatized, instead of helping them.*" (Focus group 1, Participant 4). Lastly, some thoughts regarding addressing stigma concerned educating the community. "*More education needs to be disseminated to the public and to the cultures on how this disease, you know, is transmitted. So right now, the stigma is still there though.*" (Focus group 1, Participant 3).

Overall, it was thought that there was silence around HIV/AIDS. Most people felt that they did not know how many people in the Haitian community might be living with HIV. "*I don't know how to truthfully answer that question because I really don't know. Mm, because this, again, this is not something that we talk about ever.*" (Focus group 2, Participant 15). "*I feel there is a kind of hypocrisy in this world. Why? Because the bible clearly shows us there is no sex before marriage. And*

*if we want to understand…sexually transmitted diseases, we see that they are considered as taboo. And while they are ravaging the church there isn't really much talk about it."* (Focus group 3, Participant 27).

There was an element of fear regarding HIV/AIDS. *"As a Haitian I felt nervous. We are all afraid of AIDS. All of us Haitians we are afraid of AIDS. I am not going to lie to anyone, we as Haitians once we hear someone has it, we are all afraid."* (Focus group 2, Participant 10). *"When you hear someone has AIDS you not only stay away from the person, but you also hope that it will never happen to you."* (Focus group 3, Participant 25). There was also an association between HIV/AIDS and death. *"HIV is the one of the most dangerous diseases in the world each, everyone has to prevent from this disease because it, it can cause death like, it's a long disease"* (Focus group 2, Participant 12). Some of the fear was attributed to lack of information. *"It's to show you how the lack of information can make you scared to the point where you can be convinced that you may have the disease."* (Focus group 3, Participant 24).

In addition, some expressed a belief that others thought that the cause of HIV as an illness was secondary to a curse. *"Voodoo. They don't really have it. So, they don't even want to treat it. Someone gave it to them."* (Focus group 1, Participant 5). There was also mention of an AIDS potion. *"Some Haitians think, we can't ignore this, AIDS is a potion that's given to someone."* (Focus group 2, Participant 10). Furthermore, there were additional misconceptions regarding how HIV is spread as well as painting people living with HIV in a negative light. *"We sometimes hear about infected people pricking their fingers and dipping it into food that they are preparing or in juices that others will drink. Sometimes people who work in restaurant are infected and their spitefulness make them kill a lot of other people because these are other ways people can be infected as well."* (Focus group 3, Participant 24).

### Thoughts regarding PrEP

Many of the participants were unaware of the existence of regarding PrEP. *"No, I have never heard of it. I think I heard about it once, but I am not familiar with it."* (Focus group 3, Participant 24). *"I don't think I've heard a lot of information about this. No."* (Focus group 3, Participant 21). Some felt that PrEP would be beneficial. *"So, I think this is like a, a great step forward."* (Focus group 2, Participant 15).

### Barriers to PrEP

For those who are aware of PrEP, there may be societal stigma regarding taking the medication. *"Anyway, this will look bad on her. This will look bad on her if she asks for it. One can arrive at many conclusions. They may think maybe she has other partners. Once someone makes that kind of a request, as a human being you will tend to [judge]"* (Focus group 1, Participant 4). There was also fear with regard to HIV testing. *"You have palpitations when you're taking that test regardless right even though you think you're like good, whatever um but obtaining the results is also a very anxious moment as well and some people may not want anyone to know that information".* (Focus group 1, Participant 2).

It was also thought that by promoting PrEP, one would be encouraging others to be careless in their sexual encounters. *"I believe also having preventative medicine for HIV is promoting unprotected sex.* (Focus group 2, Participant 15).

Another barrier to PrEP for the Haitian community has been the advertising. *"These commercials. And I think that's a mistake. Because in the commercials that I see with people that are saying "I am taking so and so to prevent…" it's mostly homosexual males. With that being said, people are going to feel like, ohh that's for that group".* (Focus group 2, Participant 10). Viewers may interpret this advertising campaign as a suggestion that PrEP is exclusively useful for sexual minorities.

Furthermore, another potential barrier to PrEP may be mistrust created by rushed clinical visits and inadequate communication. *"We're trying to get you in and out and not really, if you don't have an advocate for you at these places or you're not really well educated, That's, that's where the mistrust, a lot of like, oh no, they're just going to send me to get more medication."*(Focus group 2, Participant 11). *"One of the other factors that we can say causes a lack of trust is the*

*side effects. You can take medication for one disease and the side effect may bring about 4 to 5 other diseases. A lot of times we are not provided the information, we must do some research and discover these facts.*" (Focus group 3, Participant 21).

### Facilitators for PrEP

PrEP could potentially be helpful for those protecting themselves from unfaithful spouses. "*You know some people may be religious, the person may be a Christian, and her husband sleeps around, and she may not want to be exposed by her husband and she says, "Let me take this medication in case my husband catches something.*" (Focus group 1, Participant 7). Another potential facilitator may be that some Haitians would prefer not to wear condoms. "*The Haitian men already don't like to wear condoms anyway. So, it's a free for all.*" (Focus group 2, Participant 13).

Trust also seemed to play an important role. "*I know some churches have medical providers in the church. They usually have like a team, a medical team um I think if there's some sense of trust within you know people on that team, someone can confide in a member on there.*" (Focus group 1, Participant 5).

### Role of church in PrEP delivery

Respondents felt that the church should play an active role in education and prevention regarding disease. (Focus group 1, Participant 4) The church leaders felt that health fairs and general meetings could be held to promote PrEP. "*In educational activities and health fairs that we can organize.*" (Focus group 1, Participant 3). Some thought that the church should not provide any medication but could rather provide information. "*Yeah. No, but not, not here. You know, we provide PrEP, but I think we're more open to like giving that information.*" (Focus group 2, Participant 11).

### Current services provided

A lot of the churches currently provide some health education. One had also recently participated in a study. "*We had a special program recently from the women's ministry. We informed the people in part about some of these issues; how to protect themselves.*" (Focus group 3, Participant 24). "*Um, that's a big thing at our church because we do have several, um, members who are from that industry. We have many, many nurses. Many doctors here are members. So, there's always, like, in fact, last weekend we had a health, um, fair, inviting the neighborhood. You know, people came through, they had their blood, the pressures checked. Um, I think that they do blood sugar. Yeah, blood sugar. Blood sugar. And so, it was a big thing. So, breast cancer, um, month is big here.*" (Focus group 2, Participant 15)

### Interventions to help with PrEP

Some respondents felt that other members of the Haitian community would be better at conveying the message about PrEP than the churches. A participant suggested collaborating with famous figures in the Haitian community. "*So, there's these popular Haitian figures in the community that many people listen to, like my mom used to listen to um, oh gosh, I forgot his name on the radio. Piman Bouk. There are other popular Haitian people in the community, and they can also, talk about it, and they have stuff on their radio stations and stuff like that, they can talk about it.*" (Focus group 1, Participant 5).

There was also the thought that more information should be tailored to Haitians. "*I do agree, like you don't hear like the Creole commercials. It's mostly the Spanish English commercials that are like targeting, because they feel like there's a lot of, but we are here. Haitians are here. And they got to hear too.*" (Focus group 2, Participant 11).

## Discussion

This study offers a comprehensive understanding of thoughts regarding HIV/AIDS and PrEP among Christian church leaders in the Haitian community in Miami-Dade County and surrounding areas. By delving into this issue, we uncover the

nuanced interplay of cultural, religious, and social dynamics that shape health behaviors and perceptions. This complexity is particularly evident within the contexts of church structure, stigma, misconceptions about HIV/AIDS, and attitudes towards PrEP.

Central to these findings is the observation that the churches in the Haitian communities in Miami-Dade County and surrounding areas can play an important role in reducing the number of new cases of HIV through disseminating knowledge about PrEP to their congregations. As in many other churches, the democratic structure of Haitian churches reflects broader trends in religious organizations towards participatory decision-making, specifically in matters impacting community welfare [38–39]. Indeed, in many communities, this approach is crucial for effective community engagement in health initiatives. Furthermore, the emphasis on education within these churches aligns with their historical role in providing social and health education, often as pivotal community anchors [40–41].

The persistent stigma surrounding HIV/AIDS, rooted in historical misconceptions about the disease's origin, underscores the long-lasting impact that epidemiological narratives can have [42–44]. This stigma, extending into the healthcare sector, echoes challenges among healthcare professionals that impact patient care and HIV management [45]. Compounding this is the silence and fear around HIV/AIDS, prevalent in this study's findings, mirror global trends where HIV is shrouded in secrecy and fear, often fueled by misinformation and historical associations with mortality [46]. Consequently, the lack of open discourse on sexual health and HIV in the Haitian community is a significant barrier to HIV prevention and education, and culturally sensitive approaches to public health messaging are critically needed to overcome these barriers.

Moreover, these findings indicate that the belief in supernatural etiologies of HIV aligns with literature suggesting that in many cultures, diseases like HIV are often attributed to spiritual or supernatural forces [47–48]. One study suggests that men who believe that HIV can be spread through witchcraft were less likely to use protection [48]. This cultural perspective can significantly impact the community's approach to health and illness, often posing challenges to implementing conventional medical interventions.

This study reveals a considerable lack of PrEP awareness in the Haitian community. This finding echoes broader trends in HIV prevention efforts. For example, research across different populations, including African American women [49–50], adolescents and young adults in South Africa [51], consistently shows a lack of awareness about PrEP. These findings highlight the need for targeted educational campaigns that are linguistically and culturally appropriate.

In addressing barriers to PrEP uptake, it must be acknowledged that church leaders may find it difficult to educate the community on PrEP and other preventative measures, because it alludes to a lifestyle outside of their teachings. The perceived stigma associated with PrEP is significant. Such perceived stigma and fear of HIV testing are important barriers to PrEP uptake, particularly among women and in religious communities [52]. These barriers also exist, where practical considerations for PrEP implementation are important [53]. Fear of stigmatization is a key barrier to HIV testing, for example in South Africa, highlighting the need for interventions to reduce HIV-related stigma [54].

The belief that promoting PrEP equates to promoting unprotected sex found in this study is a common misconception previously described by others that needs to be addressed in community education efforts [55]. Additionally, the skewed perception of PrEP's target audience, due to advertising predominantly featuring same-sex couples, is another barrier. This trend reflects a broader issue in public health messaging where minority groups often feel underrepresented or misrepresented [56]. Moreover, the mistrust in the medical system, fueled by rushed clinical visits and inadequate communication about side effects, is another critical barrier to address [55].

This study provides insight into potential PrEP facilitators. PrEP as a measure for individuals with sexual partners outside of marriage is a significant facilitator. This perspective aligns with studies suggesting that PrEP is perceived as a tool for maintaining personal health autonomy [56–59]. The preference for non-condom use among some Haitian men indicates a potential area where PrEP could be beneficial [60]. Furthermore, the role of trust in medical professionals, particularly those affiliated with churches, is a notable facilitator for PrEP. Engaging church-affiliated medical teams in PrEP education could leverage existing trust relationships to promote PrEP uptake [61].

Moreover, the potential role of churches in PrEP education and promotion reflects the churches' influential position in many communities [62]. Organizing health fairs and educational meetings can make churches effective platforms for PrEP awareness, leveraging their role as a trusted community institution. As observed in this study, the existing framework of health education in churches provides a ready platform for integrating PrEP education. Studies have shown that health initiatives integrated into church programs can effectively reach wide community segments [60–61].

Lastly, the suggestion to collaborate with popular Haitian figures in the community for PrEP advocacy mirrors strategies used in other public health campaigns [63]. Also, the need for information tailored to the Haitian community, including Creole language resources, is a critical to effective health communication. This study has several strengths including utilizing community partnerships to conduct the study and having a bilingual English/Haitian Creole team.

These findings reinforce the importance of integrating cultural relevance into public health interventions and research. For communities like the Haitian diaspora, where language, spirituality, and historical experience deeply shape health perceptions, culturally aligned strategies are essential for building trust and achieving behavior change [62,64]. This includes developing interventions in Haitian Creole, collaborating with respected faith leaders, and acknowledging the community's unique experiences with stigma and marginalization. Culturally relevant approaches not only improve the accessibility and acceptability of health information but also help narrow persistent gaps in care and outcomes [65]. By aligning prevention efforts like PrEP education with the values and lived realities of the Haitian community, public health programs can be more effective in reducing disparities and advancing health equity.

## Limitations

This study has some limitations. Given that only three religious denominations were in this study, the findings may not be generalizable to the population. More studies exploring the role of PrEP delivery in churches and other religious settings will be important. However, as the voices/views captured are from Haitians/ Haitian Americans who are leaders of churches, the findings may not represent the Haitian community as a whole. In addition, about half of the sample was a highly educated group, with most individuals earning $50,000 or more per year. It would be important to obtain further information from individuals who are not leaders and have a lower socio-economic background.

## Conclusion

Overall, this study underscores the potential of leveraging existing community structures and preferences to facilitate PrEP uptake in the Haitian community. It is crucial that churches and trusted medical professionals collaborate with influential community figures and employ culturally relevant messaging. These findings suggest a need for community-specific, culturally relevant approaches in PrEP education and promotion, highlighting the importance of understanding and integrating community dynamics in public health interventions. Effective implementation of these strategies could significantly improve PrEP awareness and acceptance in the Haitian community.

## Supporting information

**S1 Appendix. Qualitative Focus Group Guide for Church Leaders.**
(DOCX)

**S1 File. Codebook.**
(XLSX)

## Acknowledgments

The team would like to thank the study participants, our church partners, and April Mann from the University of Miami Writing Center. In addition, we would like to thank the study funder.

## Author contributions

**Conceptualization:** Candice A. Sternberg, Allan Rodriguez, Sannisha K. Dale, Maria L. Alcaide.

**Data curation:** Candice A. Sternberg.

**Formal analysis:** Candice A. Sternberg, Maurice J. Chery, Maika Beauvoir.

**Funding acquisition:** Candice A. Sternberg.

**Project administration:** Maurice J. Chery, Maika Beauvoir, Pepita Jean.

**Writing – original draft:** Candice A. Sternberg, Maurice J. Chery, Maika Beauvoir, Pepita Jean, Dominique Guillaume, Joelle-Ann Joseph, Regine Thermy Jean Baptiste, Tyra Montour, Sannisha K. Dale, Maria L. Alcaide.

**Writing – review & editing:** Candice A. Sternberg, Maika Beauvoir, Valeria Botero, Anjalie Geffrard, Chantal Napoleon, Terese Gelin, Allan Rodriguez, John F. P. Bridges, Christopher J Hoffmann, Guerda Nicolas, Sannisha K. Dale, Maria L. Alcaide.

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
