## [Decision Letter · Decision Letter 0]

2 Jul 2025

Dear Dr. Sternberg,

Thank you for submitting your manuscript to PLOS ONE. After careful consideration, we feel that it has merit but does not fully meet PLOS ONE’s publication criteria as it currently stands. Therefore, we invite you to submit a revised version of the manuscript that addresses the points raised during the review process.

We look forward to receiving your revised manuscript.

Kind regards,

Awatif Abid Al-Judaibi, PhD

Academic Editor

PLOS ONE

This work was supported by the National Institute on Minority Health and Health Disparities, award number P50MD017347-02S1. Dr. Sternberg has also been supported by the University of Miami’s Center for HIV and Research in Mental Health (CHARM; P30MH116867 (Developmental-AIDS Research Center), P30MH133399 (AIDS Research Center) [National Institute of Mental Health]) the University of Miami CTSI through the National Center for Advancing Translational Sciences, award number KL2TR002737.

Reviewers' comments:

Reviewer's Responses to Questions

**Comments to the Author**

1. Is the manuscript technically sound, and do the data support the conclusions?

Reviewer #1: Yes

Reviewer #2: Partly

2. Has the statistical analysis been performed appropriately and rigorously?

Reviewer #1: Yes

Reviewer #2: N/A

3. Have the authors made all data underlying the findings in their manuscript fully available?

Reviewer #1: Yes

Reviewer #2: Yes

4. Is the manuscript presented in an intelligible fashion and written in standard English?

Reviewer #1: Yes

Reviewer #2: Yes

Reviewer #1: Good afternoon,

This article addresses a highly relevant topic with an appropriate methodological approach and presents significant findings that can help improve public health outcomes for Haitian and Haitians and Haitian Americans. With some minor adjustments, the manuscript could be ready for publication.

The study is technically sound, and the language is clear, professional, and accessible. It uses a qualitative design (focus groups), which is well-suited to its aim of exploring perceptions, beliefs, and cultural barriers regarding PrEP.

However, I believe it would be helpful for the article to more clearly explain how the findings can guide future interventions, clinical practices or public policies targeted at this specific population, although limitations are acknowledged in the discussion these results could serve as a starting point for exploring similar issues in other communities.

While the article’s goal is to dismantle stigma and better understand cultural barriers, the emphasis on prevalence rates—if not carefully contextualized—might unintentionally reinforce existing stereotypes about Haitians and HIV.

Suggest revising the lines (94,95 and 96) to avoid possible misunderstandings or biased interpretations, what could be misinterpreted by readers unfamiliar with the historical stigma linked to this group.

Currently, I have no further suggestions.

Thank you very much.

Reviewer #2: 1. Abstract

The type of study and design must be indicated.

2. Ethics Statement

Why wasn't a signed informed consent form used for the whole study? It's striking that the patients gave verbal consent to participate in the research but didn't sign any documents.

3. Methods

Recruitment

It is not indicated how the sample size was established.

Privacy and access to research data

It is not indicated how privacy and access to research data were protected. This should be mentioned.

Data Analysis

The current description of the data analysis process is insufficient for a qualitative study, as it lacks critical details necessary to assess methodological rigor and transparency. Specifically, the manuscript does not indicate the type of qualitative analysis conducted (e.g., thematic analysis, grounded theory), whether an inductive, deductive, or mixed coding approach was used (the summary indicates that it was inductive), or how discrepancies between coders were resolved. The process of codebook development is briefly mentioned, but no information is provided regarding its structure, validation, or use across the dataset. Moreover, the manuscript does not address how data saturation was assessed or what strategies were employed to ensure trustworthiness (e.g., triangulation, reflexivity, audit trail, or member checking). It is recommended that the authors expand this section by clearly outlining the analytic approach, coding procedures, software use, criteria for saturation, and strategies to enhance the credibility of findings.

4. Results

Although six major themes or categories of analysis were explored, this section does not indicate whether there were emerging categories, as this information is also relevant as it provides new insights that enrich the study, especially since a semi-structured approach was used for the focus group.

5. Discussion

This is a well-done section, but I think it would be improved if the importance of cultural relevance in health interventions were explored in more depth. This allows for better shaping of health strategies to achieve better outcomes for people's health and well-being, as well as helping to narrow the gaps in access to health information and healthcare.

**Do you want your identity to be public for this peer review?** For information about this choice, including consent withdrawal, please see our Privacy Policy

Reviewer #1: **Yes: ** Yenifer Lizeth Gañan Rojas

Reviewer #2: **Yes: ** Katya Cuadros-Carlesi

---

## [Author Response · Author response to Decision Letter 1]

22 Jul 2025

Reviewer #1

Yenifer Lizeth Gañan Rojas

Comments to Author:

This article addresses a highly relevant topic with an appropriate methodological approach and presents significant findings that can help improve public health outcomes for Haitian and Haitians and Haitian Americans. With some minor adjustments, the manuscript could be ready for publication. The study is technically sound, and the language is clear, professional, and accessible. It uses a qualitative design (focus groups), which is well-suited to its aim of exploring perceptions, beliefs, and cultural barriers regarding PrEP.

1) However, I believe it would be helpful for the article to more clearly explain how the findings can guide future interventions, clinical practices or public policies targeted at this specific population, although limitations are acknowledged in the discussion these results could serve as a starting point for exploring similar issues in other communities.

a) Thank you for your comment, in lines 472-482 we have added further discussions on how the findings can guide future interventions, clinical practices or public policies targeted at this specific population. “These findings reinforce the importance of integrating cultural relevance into public health interventions and research. For communities like the Haitian diaspora, where language, spirituality, and historical experience deeply shape health perceptions, culturally aligned strategies are essential for building trust and achieving behavior change (Allen et al., 2013; Pardo et al., 2021). This includes developing interventions in Haitian Creole, collaborating with respected faith leaders, and acknowledging the community’s unique experiences with stigma and marginalization. Culturally relevant approaches not only improve the accessibility and acceptability of health information but also help narrow persistent gaps in care and outcomes (Cipita et al., 2024). By aligning prevention efforts like PrEP education with the values and lived realities of the Haitian community, public health programs can be more effective in reducing disparities and advancing health equity.”

2) While the article’s goal is to dismantle stigma and better understand cultural barriers, the emphasis on prevalence rates—if not carefully contextualized—might unintentionally reinforce existing stereotypes about Haitians and HIV. Suggest revising the lines (94,95 and 96) to avoid possible misunderstandings or biased interpretations, what could be misinterpreted by readers unfamiliar with the historical stigma linked to this group.

a) Thank you for bringing this to our attention. We believe this is a very important point. In lines 95-98, we revised the sentences to “Moreover, Black, or African American individuals remain disproportionately affected by HIV largely due to structural factors and marginalization (CDC, 2021; Saunt et al., 2025). In 2021, the Center for Disease Control (CDC) data reports showed that Black or African American individuals, made up 40% of new HIV diagnoses in the U.S. (CDC, 2021).”

Reviewer #2

Katya Cuadros-Carlesi

Comments to Author:

1) Abstract- The type of study and design must be indicated.

a) Thank you for your comment. We’ve updated the abstract to include “In this qualitative study, focus groups were conducted with Haitian church leaders using snowball sampling.” This change can be found in line 45.

2) Ethics Statement- Why wasn't a signed informed consent form used for the whole study? It's striking that the patients gave verbal consent to participate in the research but didn't sign any documents.

a) Thank you for your inquiry, given the sensitive nature of HIV- related discussions regarding Haitians with church leaders, this was a confidential minimal risk study. The University of Miami IRB waived the requirements to obtain sign consent. We added this in lines 210-213.

3) Methods

Recruitment- It is not indicated how the sample size was established.

Privacy and access to research data It is not indicated how privacy and access to research data were protected. This should be mentioned.

a) Thank you for your comment, we used standard focus group methods to establish our sample size. Typically focus groups are recommended to have 6 to 10 individuals, but some guidelines and research reports note that groups as small as 4 and as large as 12 can be productive. We added to the manuscript in lines 207-209 “The sample size was established using published scientific research protocols in which the recommended number of participants are between 4-12 individuals per focus group (Teufel-Shone et al., 2010).”

In lines 191-195 we describe how privacy and access to research data were protected by stating “To protect research data and ensure privacy, all participant data was stored in a secured, password protected University of Miami approved cloud platform (Box) and data collection program (RedCap®- Research Electronic Data Capture). All study staff received training regarding participant privacy. A link between ID number and participant’s name were kept in a separate file and password protected.”

Data analysis- The current description of the data analysis process is insufficient for a qualitative study, as it lacks critical details necessary to assess methodological rigor and transparency. Specifically, the manuscript does not indicate the type of qualitative analysis conducted (e.g., thematic analysis, grounded theory), whether an inductive, deductive, or mixed coding approach was used (the summary indicates that it was inductive), or how discrepancies between coders were resolved. The process of codebook development is briefly mentioned, but no information is provided regarding its structure, validation, or use across the dataset. Moreover, the manuscript does not address how data saturation was assessed or what strategies were employed to ensure trustworthiness (e.g., triangulation, reflexivity, audit trail, or member checking). It is recommended that the authors expand this section by clearly outlining the analytic approach, coding procedures, software use, criteria for saturation, and strategies to enhance the credibility of findings.

a) In line 239-240we added “Thematic analysis was conducted to analyze data utilizing an inductive approach (Thomas et.al, 2008).” We also added how discrepancies between coders were resolved in lines 242-244 “The coders met two times to discuss their codes, emergent themes, and resolve discrepancies through collective consensus. Once consensus was reached, the final coding structure was reviewed by the PI.”

To elaborate on how we ensured trustworthiness in lines 248-251 we added “Study team members conducted member checking by presenting results to the churches to ensure interpretations accurately reflect the participants' perspectives (Mckim, 2023). Church leaders provided feedback on the results and shared their thoughts on the project.”

4) Although six major themes or categories of analysis were explored, this section does not indicate whether there were emerging categories, as this information is also relevant as it provides new insights that enrich the study, especially since a semi-structured approach was used for the focus group.

a) Thank you for your comment. In line 239, we added” Thematic analysis was conducted to analyze data utilizing an inductive approach.” Given the inductive approach, all themes were emerging.

5) Discussion- This is a well-done section, but I think it would be improved if the importance of cultural relevance in health interventions were explored in more depth. This allows for better shaping of health strategies to achieve better outcomes for people's health and well-being, as well as helping to narrow the gaps in access to health information and healthcare.

a) Thank you for your insightful comment. In response, we have added a new paragraph in the discussion section (lines 472–482) that expands on the importance of cultural relevance in health interventions, emphasizing its role in improving health outcomes and reducing disparities in access to information and care. “These findings reinforce the importance of integrating cultural relevance into public health interventions and research. For communities like the Haitian diaspora, where language, spirituality, and historical experience deeply shape health perceptions, culturally aligned strategies are essential for building trust and achieving behavior change (Allen et al., 2013); Pardo et al., 2021). This includes developing interventions in Haitian Creole, collaborating with respected faith leaders, and acknowledging the community’s unique experiences with stigma and marginalization. Culturally relevant approaches not only improve the accessibility and acceptability of health information but also help narrow persistent gaps in care and outcomes (Cipita et al., 2024). By aligning prevention efforts like PrEP education with the values and lived realities of the Haitian community, public health programs can be more effective in reducing disparities and advancing health equity.”

---

## [Decision Letter · Decision Letter 1]

7 Aug 2025

Bon bagay (good stuff):  A faith-based outlook on biomedical prevention among Haitians and Haitian Americans

PONE-D-25-27087R1

Dear Dr. Candice Aurelus Sternberg,

We’re pleased to inform you that your manuscript has been judged scientifically suitable for publication and will be formally accepted for publication once it meets all outstanding technical requirements.

Kind regards,

Awatif Abid Al-Judaibi, PhD

Academic Editor

PLOS ONE

Reviewers' comments:

Reviewer's Responses to Questions

**Comments to the Author**

Reviewer #1: All comments have been addressed

Reviewer #2: All comments have been addressed

2. Is the manuscript technically sound, and do the data support the conclusions?

Reviewer #1: Yes

Reviewer #2: Yes

3. Has the statistical analysis been performed appropriately and rigorously?

Reviewer #1: Yes

Reviewer #2: N/A

4. Have the authors made all data underlying the findings in their manuscript fully available?

Reviewer #1: Yes

Reviewer #2: Yes

5. Is the manuscript presented in an intelligible fashion and written in standard English?

Reviewer #1: Yes

Reviewer #2: Yes

Reviewer #1: This article addresses a highly relevant topic from a culturally grounded perspective: HIV prevention through PrEP in Haitian communities in South Florida, with Christian churches serving as a central point of engagement. It is a well-designed qualitative study, with strong methodological execution, rigorous analysis, and a well-developed discussion, offering concrete contributions to public health and the development of culturally responsive interventions.

The manuscript represents a valuable and original contribution at the intersection of public health, religion, and migration. It explores a historically stigmatized population from a respectful and empowering perspective, integrating key cultural elements essential to the effectiveness of PrEP interventions.

I recommend approval with minor revisions. The article is conceptually sound, well written, and presents relevant and actionable findings. The suggested improvements do not compromise the core structure of the study and can be readily addressed in a revised version of the manuscript.

Reviewer #2: After the proofreading process, the manuscript is well-structured and error-free. I hope the authors continue to deepen their knowledge on the topic, thereby developing better strategies to reduce HIV infection among the Haitian population.

**Do you want your identity to be public for this peer review?** For information about this choice, including consent withdrawal, please see our Privacy Policy

Reviewer #1: No

Reviewer #2: **Yes: ** Katya Cuadros-Carlesi

---

## [Editor Report · Acceptance letter]

PONE-D-25-27087R1

PLOS ONE

Dear Dr. Sternberg,

I'm pleased to inform you that your manuscript has been deemed suitable for publication in PLOS ONE. Congratulations! Your manuscript is now being handed over to our production team.

Kind regards,

on behalf of

Professor Awatif Abid Al-Judaibi

Academic Editor

PLOS ONE